

# Retrospective evaluation of the morphometric properties of intact maxillary sinus using cone-beam computed tomography for sex estimation in an Indian population

Vrushali Raosaheb Ghavate[1,*], Ajinkya M. Pawar[1,*], Jatin Atram[1], Vineet Vinay[2], Dian Agustin Wahjuningrum[3], Alexander Maniangat Luke[4,5] and Nader Nabil Rezallah[6]

[1] Conservative Dentistry and Endodontics, Nair Hospital Dental College, Mumbai, Maharashtra, India
[2] Department of Public Health Dentistry, Sinhgad Dental College and Hospital, Pune, Maharashtra, India
[3] Department of Conservative Dentistry, Faculty of Dental Medicine, Universitas Airlangga, Surabaya City, East Java, Indonesia
[4] Department of Clinical Science, College of Dentistry, Ajman University, Al-Jurf, Ajman, United Arab Emirates
[5] Center of Medical and Bio-allied Health Sciences Research, Ajman University, Al-Jurf, Ajman, United Arab Emirates
[6] Department of Oral and Maxillofacial Radiology, City University Ajman, Sheikh Ammar Road, Ajman, United Arab Emirates
[*] These authors contributed equally to this work.

Corresponding authors
Ajinkya M. Pawar,
ajinkya@drpawars.com
Dian Agustin Wahjuningrum, dian-agustin-w@fkg.unair.ac.id

## ABSTRACT

**Background.** Sex estimation is crucial to forensic examinations. In order to estimate sex, intact bones are used if the majority of bones are severely deformed and recovered in fragments. This study aims to analyze sexual dimorphism in intact maxillary sinuses using CBCT scanning to evaluate morphometric properties for sex identification.

**Methods.** A total of 318 subjects, consisting of 159 males and 159 females, aged between 20 and 60 years without sinus pathology were included in this diagnostic, retrospective cross-sectional study. Bilateral measurements of the volume, height, width, and length of the maxillary sinuses were obtained and compared to evaluate the differences between sexes. Subsequently, a descriptive analysis using mean and standard deviation was performed, followed by a comparison between sexes with a p-value being less than 0.05 and Student's t-test. Finally, a discriminant analysis was performed separately for the right and left maxillary sinuses.

**Results.** Males and females showed statistically significant variations in the length, width, and volume of the maxillary sinuses. Specifically, on the right side, males had longer maxillary sinuses than females ($t = 5.6203$, $p < 0.0001$). Meanwhile, on the left side, females had wider maxillary sinuses than males ($t = 8.621$, $plt\,0.0001$). In addition, males had greater volumes of maxillary sinuses on the right ($t = 6.373$, $p < 0.0001$) and left ($t = 3.091$, $p < 0.0001$) sides than females. The results of the discriminant analysis showed that the left width parameter had the highest accuracy of sex estimation (74.21%), followed by the Right Length (70.07%) and left volume (66.66%) parameters. The left height parameter had the lowest accuracy of sex estimation (49.37%).
**Conclusion**. In forensic odontology, the volume of maxillary sinus can serve as a valid radiographic indicator of sex estimation.

## INTRODUCTION

Personal identification is crucial to forensic investigations, especially in the aftermath of natural disasters such as traffic accidents, fire accidents, plane crashes, and other mass tragedies (*Krishan et al., 2012*). In these cases, forensic science plays a critical role in collecting and analyzing information, as well as identifying suspects, making it an essential part of the criminal justice system (*Pittayapat et al., 2012*). The Law of Individuality in forensic science states that every material, natural or man-made, has unique features that cannot be replicated in any other substance, either naturally or artificially (*Cattaneo, 2007*). This law provides the basis for distinguishing individuals by their characteristics. Forensic scientists use various techniques, such as DNA profiling, fingerprint analysis, and dental records, to evaluate and compare forensic evidence (*Manjunath et al., 2011*). Therefore, the combination of postmortem evidence with antemortem data is essential to establish the identity of an individual. Dental records are particularly important in the identification of individuals through forensic odontology (*Pretty & Sweet, 2001*).

Sex estimation is crucial to forensic examinations. This is especially true when only skeletal remains are available for examination. For example, the pelvis, skull, sternum, mastoid process, epiphyses, scapula, and metacarpal bones have all been used to estimate sex (*Bidmos et al., 2021*). A prominent forensic anthropologist developed a rating system to ensure the accuracy of sex estimation using various bones, with the pelvis being the most accurate at 95%, the skull at 90%, and the long bones at 80% (*Math et al., 2014*). However, these bones are frequently recovered in fragments and disarray, making it more difficult to identify them (*Najem et al., 2020*; *Ricardo et al., 2022*). Therefore, researchers have turned to denser bones that can be recovered in a more intact state, which aids in sex identification (*Dangore-Khasbage & Bhowate, 2018*; *Nunes Rocha, Dietrichkeit Pereira & Alves da Silva, 2021*).

Forensic odontology is a specialized field within forensic science that uses dental evidence to aid in criminal investigations, identify missing persons, and provide evidence in court proceedings. One of the techniques used in forensic odontology is the morphometric analysis of maxillary sinus, which can provide important information about the identity and lifestyle of an individual (*Introna et al., 2008*). The size and shape of the maxillary sinus can vary between individuals and be influenced by factors such as age, genetics, and sex. Therefore, the morphometric analysis of the maxillary sinus has the potential to be a valuable tool in forensic examinations (*Reda et al., 2021*; *Shrestha et al., 2021*). It can also provide information about the medical and dental history of the individual, which can be helpful in criminal investigations (*Pramod, Marya & Sharma, 2012*).

Several studies have demonstrated the potential applications of the morphometric analysis of maxillary sinus in forensic examinations. For example, in a sample of Italian adults, the morphometric analysis of the maxillary sinus showed that the volume of the maxillary sinus correlated positively with age and negatively with body mass index (BMI) (*Saccucci et al., 2015*). Another morphometric analysis of maxillary sinus in adults revealed that males had a significantly greater volume of the maxillary sinus than females (*Kandel, Sharma & Sah, 2020*). Interestingly, the volume of maxillary sinus was found to be significantly greater in individuals who engaged in intense physical labor, suggesting a potential correlation between the volume of maxillary sinus and occupational activity (*Whyte & Boeddinghaus, 2019*; *Pérez Sayáns et al., 2020*). In addition to volume, it was revealed that the length and width of maxillary sinus can be used to accurately differentiate between males and females (*Prabhat et al., 2016*).

The maxillary sinus is a cavity that is located in the maxilla, above the roots of the upper molars. It is a unique anatomical structure that can be used for forensic identification due to its ability to resist damage from various factors, including trauma, decomposition, and postmortem. This resistance to damage allows the maxillary sinus to remain intact even in cases of severe decomposition or trauma, making it a useful tool for sex estimation. In forensic examinations, sex can be estimated based on studies that show differences in the size and shape of maxillary sinuses between males and females. In cases of other approaches, such as DNA analysis, are not feasible or have failed to provide solid findings (*Xavier, Dias Terada & Da Silva, 2015*), the morphometric analysis of maxillary sinus can be a helpful tool for sex estimation.

In this context, it is important to distinguish between identification and biological profile estimation. Identification is the process of estimating the sex of an individual based on predetermined criteria. Meanwhile, biological profile estimation involves a broader evaluation that takes into account various biological traits, including but not limited to sex, to create a detailed profile of the individual (*Spradley, 2016*). In addition to sex estimation, the maxillary sinus can be used for personal identification, for instance, by comparing the unique shape and size of the maxillary sinus to radiographs or other images. This technique has been successfully applied in forensic investigations to identify victims of mass disasters or in cases where the body is severely decomposed (*Ricardo et al., 2022*).

Non-invasive imaging techniques such as computed tomography (CT) and magnetic resonance imaging (MRI) produce images that can be analyzed for linear and volumetric measurements. These measurements are generated using the images obtained and software provided by the imaging equipment manufacturer or third-party applications developed for this purpose. CT scanning can accurately calculate the volume and dimensions of maxillary sinus, making it a valuable tool for sex estimation in forensic examinations (*Ricardo et al., 2022*). Cone-beam computed tomography (CBCT) is a modified form of the traditional CT that can generate high-quality diagnostic images of the oral and maxillofacial regions with lower radiation doses and processing times, making it a safer and more convenient option for patients (*Neves et al., 2014*). Contrast-enhanced CBCT can generate multiple images of the maxillofacial complex from a single scan, including sagittal, coronal, and axial images, as well as a 3D reconstruction of the bony skeleton. As a result, CBCT is considered the

gold standard for the oral and maxillofacial imaging due to its high diagnostic accuracy (*Scarfe et al., 2012*).

Given the above context, this study aims to evaluate the accuracy of different characteristics of maxillary sinuses in identifying sexual dimorphism.

## MATERIAL AND METHODS

This retrospective study was conducted in accordance with the Strengthening the Reporting of Observational Studies in Epidemiology (STROBE) guidelines and approved by the institutional ethics committee of Nair Hospital Dental College on September 14, 2022 with a certificate number NHDC/ND EC-177/CONS/ND107/2022. Because of the retrospective nature of this study, the ethics committee waived the requirement for informed consent. This study involved a retrospective analysis of 318 CBCT scans of an Indian population obtained from the Oral Radiology department. These scans were originally obtained as part of a diagnostic regimen for a different treatment plan unrelated to the current study. CBCT imaging is a high-quality diagnostic imaging technique that generates accurate three-dimensional images of the oral and maxillofacial regions, making it a valuable tool for dental and medical practitioners. In this study, this imaging technique was used to evaluate the accuracy of different characteristics of maxillary sinuses in estimating sexual dimorphism.

The CBCT scans used in this study were taken with a ProMax 3D Mid unit (Planmeca, Helsinki, Finland). The scanning parameters were set to a tube voltage of 90 kVp, a tube current of 6 mA, and an exposure time of 6.12 s. The sample size for the study was determined using the OpenEpi software, which estimated the required number of scans with a 95% confidence interval, a 5% alpha error probability, and an 80% power of study. With the expected minimum odds ratio of 0.42, the total sample size was estimated to be 318 CBCT scans. To be eligible for this study, patients had to be over 20 years old and had no sinus pathology, such as acute or chronic sinusitis, mucositis, osteomyelitis, radicular cyst, antral polyp, maxillary sinus fracture, exostosis, endostosis, or lethal midline granuloma. The CBCT scans were analyzed using the Romexis software (Planmeca, Helsinki, Finland). This software was used to measure the dimensions of the maxillary sinus, including length, width, and height. The volume was calculated using another formula by *Bangi et al. (2017)*. The formula used to calculate the volume of the maxillary sinus in this study was based on previous studies that emphasized the importance of sex estimation in forensic medicine. These studies also identified the maxillary sinus as a valid anatomical feature for sex estimation, especially when other approaches have failed to provide solid findings. In one of these studies, the researchers used CT imaging to accurately estimate the dimensions and volume of maxillary sinuses in a group of 100 patients, demonstrating the effectiveness of the method in sex estimation for forensic identification. This study used a discriminant analysis and statistical comparisons of male and female patients to demonstrate the accuracy of sex estimation using the characteristics of the maxillary sinuses.

i. Length (in (mm)), *i.e.,* the longest distance between the most anterior point and the most posterior point of the medial wall of the maxillary sinus in the axial view (see Fig. 1A).

**Figure 1  The measurements recorded for maxillary sinus using CBCT.** (A) Length measured in axial view; (B) width measured in coronal view; and (C) height measured in coronal view.

ii.  Width (in mm), *i.e.,* the longest distance between the medial wall of the sinus and the most lateral wall of the lateral process of the maxillary sinus in the coronal view, perpendicular to each other (see Fig. 1B).

iii. Height (in mm), *i.e.,* the longest distance between the lowest point of the sinus floor to the highest point of the sinus roof in the coronal view (see Fig. 1C).

iv.  Volume (in mm$^3$), *i.e.,* (length × width × height) × 0.5.

Data were recorded for the above parameters for the left and right sides, tabulated in Microsoft Excel, and subjected to further statistical analysis. The statistical analysis was performed using Statistical Product and Service Solutions (SPSS) Version 21 for Windows (SPSS Inc., Chicago, IL, USA). Data normality was tested using the Shapiro–Wilk test. Descriptive quantitative data were presented as mean and standard deviation. Sex-specific validation was performed using the Student's *t*-test, with a *p*-value of less than 0.05 with a 95% confidence interval. The discriminant analysis was also performed. An area under the curve (AUC) with a receiver operating characteristic (ROC) curve was obtained to analyze the accuracy of the parameters in estimating sexual dimorphism.

## RESULTS

To improve the validity and reliability of this study, the dimensions of maxillary sinuses were evaluated using intra- and inter-examiner statistical tests. In the intra-examiner evaluation, two examiners measured the dimensions of the same maxillary sinus. This resulted in an average intra-class correlation coefficient (ICC) value of 0.92, indicating good accuracy and precision between their measurements. In the inter-examiner evaluation, two examiners assessed the dimensions for the inter-examiner evaluation, resulting in an ICC value of 0.88, indicating good agreement and inter-observer reliability. To reduce bias and ensure impartiality during the assessment, blinding was used. The examiners were blinded to the previous measurements of the maxillary sinuses under investigation. Furthermore, to reduce systematic errors or bias, the scans were shown to the examiners in a randomized order. To account for possible variations, each examiner measured the dimensions of the maxillary sinus three times, with at least three measurements for each dimension. The
**Table 1 The comparative analysis of the maxillary sinus measurements between the right and left sides based on sex.** Asterisks (*) indicate that the values are statistically significant.

| Study variables | Sex | n | Mean | Std. dev. | t-test |
|---|---|---|---|---|---|
| Length in mm | | | | | |
| Left | Male | 159 | 33.36 | 4.66 | 0.143 |
| | Female | 159 | 32.89 | 3.17 | |
| Right | Male | 159 | 35.38* | 4.24 | $t = 5.6203$, <0.0001* |
| | Female | 159 | 32.24 | 2.23 | |
| Width in mm | | | | | |
| Left | Male | 159 | 23.41 | 3.81 | $t = 8.621$, <0.0001* |
| | Female | 159 | 27.76* | 4.13 | |
| Right | Male | 159 | 23.67 | 2.55 | 0.1244 |
| | Female | 159 | 24.85 | 2.49 | |
| Height in mm | | | | | |
| Left | Male | 159 | 28.34 | 2.87 | 0.4111 |
| | Female | 159 | 28.42 | 3.42 | |
| Right | Male | 159 | 28.30 | 2.85 | 0.1459 |
| | Female | 159 | 27.97 | 2.73 | |
| Volume in mm³ | | | | | |
| Left | Male | 159 | 13,017.89* | 2,883.68 | $t = 6.373$, <0.0001* |
| | Female | 159 | 10,938.75 | 2,915.35 | |
| Right | Male | 159 | 13,834.79* | 1,900.63 | $t = 3.091$, <0.0001* |
| | Female | 159 | 11,205.92 | 1,710.50 | |

average of these measurements resulted in a mean intra-examiner ICC value of 0.91 for all dimensions, indicating good agreement among their multiple measurements.

The results showed that of the 318 scans evaluated, the average age was 33.75 years (SD ± 23.47) for males and 34.63 years (SD ± 19.54) for females. The age distribution between males and females was not significantly different ($p = 0.358$). Table 1 presents the mean value, standard deviation, and $p$-value for each variable for both sexes. Furthermore, according to the Student's $t$-test, the right length (in mm) of the maxillary sinus in males was significantly longer than in females ($t = 5.6203$, $p < 0.0001$). Additionally, the left width (in mm) of the maxillary sinus in females was wider than in males ($t = 8.621$, $p < 0.0001$). Finally, the volumes (in mm³) of the maxillary sinus on the right ($t = 6.373$, $p < 0.0001$) and left ($t = 3.091$, $p < 0.0001$) sides in males were greater than in females.

Furthermore, the discriminant analysis revealed remarkable results regarding sex estimation based on various parameters. The left width parameter showed an accuracy rate of 74.21% in identifying sexes, while the right length parameter showed an accuracy rate of 70.07%. In addition, the left volume parameter showed an accuracy rate of 66.66% in estimating sexes. However, the left height left parameter showed the lowest accuracy of 49.37% in distinguishing between sexes (Table 2).

Moreover, the analysis of the AUC revealed significant results (Fig. 2). Cut-off points were determined for the right length, left width, and left volume parameters since their

**Table 2** Classification function coefficients and accuracy level for each parameter in determining gender.

| | Male | | Female | | Correctly identified % |
|---|---|---|---|---|---|
| | Coefficient | Constant | Coefficient | Constant | |
| **Length Left** | 2.231 | −38.011 | 2.196 | −36.862 | 56.60 |
| **Length Right** | 3.248 | −58.891 | 2.937 | −48.278 | 70.07 |
| **Width Left** | 2.892 | −47.146 | 2.507 | −35.579 | 74.21 |
| **Width Right** | 3.905 | −46.595 | 4.126 | −51.920 | 58.49 |
| **Height Left** | 3.001 | −43.080 | 3.014 | −43.438 | 49.37 |
| **Height Right** | 3.711 | −53.144 | 3.677 | −52.197 | 51.57 |
| **Volume Left** | .002 | −16.813 | .002 | −12.529 | 66.66 |
| **Volume Right** | .003 | −17.082 | .003 | −15.328 | 57.54 |

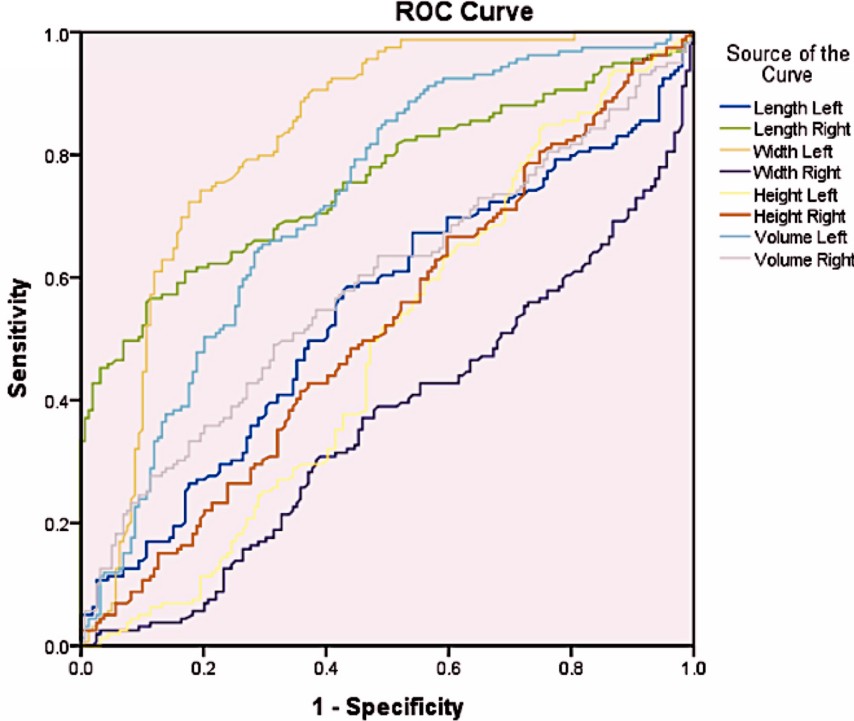

**Figure 2** ROC curve for gender determination for different parameters.

AUC values were greater than 0.7. The cut-off value for the right length parameter was 33.21, with a sensitivity of 70.4% and specificity of 61.6%, while the cut-off value for the left width parameter was 30.11, with a sensitivity of 79.9% and specificity of 70.4%. In addition, the cut-off value for the Left Volume parameter was 13,689.99, with a sensitivity of 70.4% and specificity of 61.6%. If the values were greater than or equal to the cut-off values, the sex was estimated to be male (Table 3).
**Table 3   Area under the curve for gender determination for different parameters.**

| Area Under the Curve | |
| --- | --- |
| **Test Result Variable(s)** | **Area** |
| Length Left | .547 |
| Length Right | .759 |
| Width Left | .828 |
| Width Right | .367 |
| Height Left | .487 |
| Height Right | .521 |
| Volume Left | .724 |
| Volume Right | .586 |

# DISCUSSION

In forensic investigations, sex estimation is crucial to personal identification. Sex estimation can provide important clues and reduce the need for searching for potential matches when the personal identification is unknown. This process is an essential part of forensic anthropology, which involves examining multiple skeletal structures. Sex estimation is especially important in cases where remains are badly decomposed, fragmented, or burned, and other methods of identification, such as fingerprint analysis or DNA profiling, may not be feasible. In such situations, forensic odontologists may use dental evidence to aid in sex estimation, for example, by analyzing the size and shape of maxillary sinus (*Ricardo et al., 2022*).

*Williams & Rogers (2006)* found that the distinctive characteristics of the skull and mandible can be used to determine the sex of an individual with an accuracy rate of 96%. This highlights the significance of craniofacial features in sex estimation, which is crucial to forensic anthropology and odontology. Accurate sex estimation based on skeletal features is particularly valuable in forensic investigations, where other methods of identification are impractical. For example, the use of six features in sex estimation, including the mastoid process, supraorbital ridge, skull architecture, extension of the zygomatic arch beyond the external auditory canal, nasal aperture, and gonial angle of the mandible, resulted in an accuracy of 94% (*Nagare et al., 2018*).

This study evaluates the effectiveness of using the measurements of maxillary sinus for sex estimation. Previous studies by *Urooge & Patil (2017)*, *Tambawala et al. (2016)*, *Kandel, Sharma & Sah (2020)*, *Sharma, Jehan & Kumar (2014)*, and *Sheikh et al. (2018)* have also examined different measurements of the maxillary sinus, including length, width, height, and volume, to determine their accuracy for sex estimation (*Nunes Rocha, Dietrichkeit Pereira & Alves da Silva, 2021*; *Deshpande et al., 2022*). The maxillary sinus can be affected by genetic disorders, the silent sinus syndrome, post-infectious conditions, environmental factors, and age-related variables (*Petraroli et al., 2020*). As a result, patients with maxillary sinus abnormalities were excluded from this study.

Morphometric analysis of maxillary sinus has been used as a tool for sex estimation in forensic and anthropological investigations due to its easy accessibility and visualization

using radiographic techniques (*Ekizoglu et al., 2014*). Studies have shown clear differences between males and females in the size of the maxillary sinus. It was found that males had significantly larger maxillary sinuses than females in terms of height, width, and volume (*Saccucci et al., 2015*). Additionally, it was found that males had a significantly larger capacity of the maxillary sinus than females (*Akhlaghi et al., 2017*; *Perrotti et al., 2021*; *Esfehani et al., 2023*). In this study, a statistically significant difference was observed between sexes with respect to the length of the right side of the maxillary sinus, with males having longer maxillary sinuses than females. Moreover, this study found that the width of the left side of the maxillary sinus was significantly wider in females than in males. These findings are consistent with previous studies by *Kandel, Sharma & Sah (2020)* and *Sheikh et al. (2018)*, which also reported significant differences in the length of the maxillary sinus between males and females. Furthermore, *Tambawala et al. (2016)* noted that the length of the right and left sides of the maxillary sinus was shorter in females than in males. This finding is in accordance with previous studies that found no significant differences between males and females in the dimensions of the right and left sides of the maxillary sinus on the right and left sides of an individual (*Sharma, Jehan & Kumar, 2014*; *Saccucci et al., 2015*; *Tambawala et al., 2016*; *Kandel, Sharma & Sah, 2020*).

The results of this study do not support the claims made by *Sheikh et al. (2018)* and *Deshpande et al. (2022)* that the right side of the maxillary sinus of an individual was longer and wider than the left side. *Urooge & Patil (2017)*, *Sheikh et al. (2018)*, and *Ahmed, Gataa & Mohammed (2015)* argued that the width of the left side of the maxillary sinus is the most reliable parameter for distinguishing between males and females with an accuracy rate of 61.3% (*Deshpande et al., 2022*). Contrary to these findings, however, *Tambawala et al. (2016)* and *Sharma, Jehan & Kumar (2014)* found no statistically significant differences between males and females in terms of the width of the right and left sides of the maxillary sinus.

Furthermore, the statistical analysis found no significant differences between males and females in terms of the height of the maxillary sinus on the right and left sides. This is consistent with the findings reported by *Urooge & Patil (2017)* and *Bangi et al. (2017)*. However, it is contrary to the observations made by *Deshpande et al. (2022)* that the length on the left side of the maxillary sinus was longer in males than in females, and that it can be the most reliable factor for sex estimation. In addition, the results of this study are in contradiction with the findings of *Tambawala et al. (2016)* and *Kandel, Sharma & Sah (2020)*. This study found that the measurements of the maxillary sinus in males were found to be significantly larger than in females, which is consistent with previous studies (*Sharma, Jehan & Kumar, 2014*; *Bangi et al., 2017*). However, these findings are in contrast to the observations made by *Kandel, Sharma & Sah (2020)* who reported incongruent results.

A recent systematic review evaluated the effectiveness of CBCT in forensic dentistry, specifically in sex estimation (*Nunes Rocha, Dietrichkeit Pereira & Alves da Silva, 2021*). The review included ten studies which analyze the maxillary sinus and concluded that this method has high sensitivity and specificity values for sex estimation. The review also indicated that CBCT is a non-invasive method that provides more accurate results

for measuring the maxillary sinus compared to other traditional methods (*Nunes Rocha, Dietrichkeit Pereira & Alves da Silva, 2021*).

*Teixeira et al. (2020)* assessed 420 scans of maxillary sinuses to determine their sexes. The study found that the length of the right side of the maxillary sinus was the most accurate sex determinator, with an accuracy rate of 66.9%. In contrast, this study found that the Left Width parameter was the most accurate. Meanwhile, *Gulec et al. (2020)* investigated the correlation between sex and the volume of maxillary sinus. The study found that the average volumes for the right and left maxillary sinuses were 13.173 cm$^3$ and 13.194 cm$^3$, respectively, indicating no significant differences in the volumes of the right and left maxillary sinuses. Moreover, no significant differences in the volumes of the maxillary sinuses were observed between sexes. However, this study found significant differences in the volume of the maxillary sinus between sexes. According to *Gulec et al. (2020)* males had significantly longer maxillary sinuses, resulting in an accuracy rate of 77.7% for sex estimation. Similarly, this study found statistically significant differences in the length, width, and volume of maxillary sinuses in males and females. In particular, it was found that the right side of the maxillary sinus in males were longer than the left side. In addition, the width of the maxillary sinus in females was wider than in males, while the volume of the maxillary sinus in males was significantly greater than in females. Furthermore, the discriminant analysis showed that the left width parameter was the most accurate in sex estimation with an accuracy rate of 74.21%. The right length parameter came in second with an accuracy rate of 70.07%, while the left volume parameter came in third with an accuracy rate of 66.66%. However, the left height parameter showed the least accuracy at 49.37%.

This study has demonstrated that sex-based differences in the maxillary sinus can be determined. However, the differences may be attributed to variations in the measurement techniques, sample sizes, or other factors. Increased collaboration among studies and further investigation may contribute to validating and improving of the accuracy of sex estimation using the measurements of the maxillary sinus.

The correlation between age and the volume of maxillary sinus is a topic of debate among scholars. CT imaging studies have reported varying results. Some studies suggest that the volume of maxillary sinus reduces during the second or third decade of life, while other studies, which used less accurate measurement techniques or worked with older participants, did not find any significant relationship between age and the volume of maxillary sinus. These findings are consistent with previous research, which has shown a decrease in the volume of maxillary sinus, particularly in males, after the third decade. However, to enhance the validity of these results, future research should incorporate a more diverse sample with varied age and sex distributions, as the current example primarily included young females.

Many scholars have conducted research on sexual dimorphism by analyzing the maxillary sinus using CT scans. In India, few studies with a large sample size have used CBCT scans. Therefore, this study contributes to the literature by evaluating the reliability of using the measurements of the maxillary sinus to determine sex. The main strength of this study lies in its comprehensive analysis using sophisticated and state-of-the-art software to evaluate

CBCT scans, which is considered one of the most reliable tools. However, a limitation of this study is the small and specific population, as well as the unequal distribution of males and females.

Further research should focus on broadening demographic inclusion by including various age and sex groups. Longitudinal investigations are recommended to understand age-related changes in the characteristics of the maxillary sinus. For the advancement of the discipline, it is essential to integrate CBCT with other forensic methods, validate it in forensic contexts, and investigate genetic and environmental factors. Morphometric analysis of maxillary sinus for forensic applications is not feasible without a partnership with forensic practitioners, the integration of modern imaging technology, and public education campaigns.

## CONCLUSION

The study estimated sex based on the measurements of the maxillary sinus. Males had longer maxillary sinuses on the right side as well as a greater volume of the right side of the maxillary sinuses, while females had wider maxillary sinuses on the left side. Accurate sex estimation was possible through the discriminant analysis of the measurements of the maxillary sinuses, with greater accuracy in females when using both right and left measurements. These findings have important implications for forensic investigations, particularly in cases where sex estimation is crucial. Therefore, it is important to take into account sex-specific anatomical variations when analyzing and interpreting the measurements of maxillary sinus in forensic investigations.

## ACKNOWLEDGEMENTS

The authors express their gratitude to Dr. Shivani Singh, a resident in the Department of Oral Medicine and Radiology, for her valuable insights in reviewing the CBCT scans.

### Funding

The authors received no funding for this work.

### Competing Interests

Ajinkya M. Pawar is an Academic Editor for PeerJ.

### Author Contributions

- Vrushali Raosaheb Ghavate conceived and designed the experiments, performed the experiments, prepared figures and/or tables, and approved the final draft.
- Ajinkya M. Pawar conceived and designed the experiments, performed the experiments, prepared figures and/or tables, and approved the final draft.
- Jatin Atram conceived and designed the experiments, performed the experiments, prepared figures and/or tables, and approved the final draft.

- Vineet Vinay analyzed the data, prepared figures and/or tables, authored or reviewed drafts of the article, and approved the final draft.
- Dian Agustin Wahjuningrum analyzed the data, authored or reviewed drafts of the article, and approved the final draft.
- Alexander Maniangat Luke analyzed the data, authored or reviewed drafts of the article, and approved the final draft.
- Nader Nabil Rezallah analyzed the data, authored or reviewed drafts of the article, and approved the final draft.

### Ethics

The following information was supplied relating to ethical approvals (*i.e.*, approving body and any reference numbers):

Institutional Ethics Committee of Nair Hospital Dental College

### Data Availability

The raw measurements

### Supplemental Information

Supplemental information for this article can be found online at http://dx.doi.org/10.7717/peerj.16991#supplemental-information.

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
