# Peer review of "Retrospective evaluation of the morphometric properties of intact maxillary sinus using cone-beam computed tomography for sex estimation in an Indian population"

_PeerJ, doi:10.7717/peerj.16991_

## Round 0.1 · original submission · Major Revisions

Please pay particular attention to the points raised by reviewers 1 and 4.

**Language Note:** PeerJ staff have identified that the English language needs to be improved. When you prepare your next revision, please either (i) have a colleague who is proficient in English and familiar with the subject matter review your manuscript, or (ii) contact a professional editing service to review your manuscript. PeerJ can provide language editing services - you can contact us at copyediting@peerj.com for pricing (be sure to provide your manuscript number and title). – PeerJ Staff

Reviewer 1 ·

Basic reporting

The study refers to an important subject in the area of forensic sciences. However, some points need correction and/or clarification.
1. Please note that there are initially two major conceptual issues:
a. “Gender” refers to a social issue. “Sex” referes to biological/physical characteristics. In Forensics, we deal with “sex” estimation, and not “gender”. Please replace all words "gender" with "sex" throughout the manuscript.
b. In Forensics we do not use “gender identification” nor “gender recognition”, since sex is estimated. We don't talk about sex identification because we don't have 100% accuracy. We make an estimation. Please replace all “gender identification”, “gender recognition”, “confirmation of sex”, “gender determination” and similar tems with “sex estimation”.

2. In the manuscript there are some words separated by a hyphen. Please correct. eg lines 75, 119, etc.

3. Lines 109-111: DNA analysis is an identification method (as fingerprints and dental records). It cannot be compared to sex estimation. Sex estimation is importante for estimating the biologcal profile of na individual. Please correct the frase and explain the diferences (identificaion vs biological profile estimation). Please provide the due citations/references.

4. Lines 117-118: CT scans DO NOT CALCULATE volume and dimensions. CT scans provide the images. The linear and volumetric measurements are calculated by other means (as software), that use the images to do so. Please correct.

5. Line 124: Please replace “anterior” with “sagittal”.

6. Lines 176/177: Please replace “men” with “males”.

Experimental design

1. In Material & Methods section, please mention that the sample consists on Indian population.

2. Line 153-154: The software was not used to measure the volume, that was calculated, according to the text, with the use of a formula. Please correct.

3. Lines 157-159: The correspondent image shows a coronal view, not an axial view. Please correct. Also in legend (line 481).

4. Line 162: Please provide consistent scientific justification/validation for using this formula to calculate volume. This formula was used by Banji et al. (2017), but the authors did not justify its use.
It is better to deconsider the volume data.
Currently, the calculation of the volume of the paranasal sinuses is performed by software that work with 3D reconstructed CBCT images. See studies by:

- Barros F, Fernandes CMS, Kuhnen B, Scarso Filho J, Gonçalves M, Gonçalves V, Serra MC. (2022). Three-dimensional analysis of the maxillary sinus according to sex, age, skin color, and nutritional status: A study with live Brazilian subjects using cone-beam computed tomography. Archives of Oral Biology 139: 105435. https://doi.org/10.1016/j.archoralbio.2022.105435
- Gulec M, Tassoker M, Magat G, Lale B, Ozcan S, Orhan K. (2020). Threedimensional volumetric analysis of the maxillary sinus: a cone-beam computed tomography study. Folia Morphologica, 79(3): 557–562. https://doi.org/10.5603/FM.a2019.0106.
- Bornstein MM, Ho JKC, Yeung AWK, Tanaka R, Li JQ, Jacobs R. (2019). A retrospective evaluation of factors influencing the volume of healthy maxillary sinuses based on CBCT imaging. International Journal of Periodontics and Restorative Dentistry. 39(2): 187–193. https://doi.org/10.11607/prd.3722
- Gomes AF, Gamba TO, Yamasaki MC, Groppo FC, Haiter Neto FH, Possobon R F (2019). Development and validation of a formula based on maxillary sinus measurements as a tool for sex estimation: a cone beam computed tomography study. International Journal of Legal Medicine, 133(4): 1241–1249. https://doi.org/10.1007/s00414-018-1869-6

Validity of the findings

1. To validate the findings, two examiners are necessary, as well as the performance of intra and inter-examiner statistical tests. If this was not done, it is a serious limitation of the study.

2. Lines 203-214 and 240-258: Please delete. These paragraphs are out of context, and have nothing to do with the specific issue.

3. Discussion is poor, and needs to be improved.

4. Lines 234-236: Please cite more studies.

5. In Discussion section, volume is cited only once (line 264). If robustly justified the employment of the presented formula for volume calculation, volume findings also need to be more discussed. But I suggest to disconsider the volume data, due the above-mentioned reason.

6. Please disconsider the Conclusion presented in the Abstract.

Additional comments

There are some major issues: the conceptual ones (gender vs sex, identification vs estimation) and the methodological ones (calculation of volume, only one examiner). Besides some minor presented issues.
I suggest to use a second observer, and make the due statistical intra and inter-observer validation analysis. And to disconsider volume data, or calculate volume using a software that measures the 3D reconstructed DICOM images.

Reviewer 2 ·

Basic reporting

Ok.

Experimental design

Ok.

Validity of the findings

Ok.

Additional comments

The article is well prepared and written. It is suitable for publication.

Reviewer 3 ·

Basic reporting

Query 1. Kindly elaborate or give details of the calculation of volume is derived from which mathematical formula or which shape since maxillary sinus has an uneven shape. If you have calculated it with software then kindly correct it in the article.

Query 2. Line no. 291. Kindly recheck the statement "left sided height is more common in male than females". Kindly correct it.

Query 3. Line no. 303. Kindly recheck the statement "CBCT is less invasive". Rewrite or correct the statement. CBCT is a non-invasive technique.

Experimental design

No comment

Validity of the findings

No comment

Additional comments

No comments

Reviewer 4 ·

Basic reporting

Since this article discusses the forensic benefits of Maxillary Sinus (MS) morphometry, add a comparison with some cadaveric studies of MS to distinguish the morphology between live MS and cadaveric MS.

Very minimal articles from the Indian population have been discussed in this study on the morphometry of MS. I suggest citing more Indian articles in the discussion to know more about regional changes in MS morphology in the country.
E.g.
Gupta C, Kumar S. Antony SD, Lakshmi NK. A study of morphometric evaluation of the maxillary sinuses in normal subjects using computer tomography images. Arch Med Health Sci 2014;2(01):12

Tiwari, S. T., U., S., & MM, J. (2023). Gender Determination by Measuring Maxillary Sinus Volume Using Computed Tomography. Journal of Health and Allied Sciences NU, 13(01), 064–072. https://doi.org/10.1055/s-0042-1748633

A.P. Mishra, Kuldeep Kumar, C. S. Ramesh Babu. MORPHOMETRIC STUDY OF MAXILLARY SINUSES IN NORMAL SUBJECTS BY USING COMPUTED TOMOGRAPHIC IMAGES. Int J Anat Res 2020;8(2.2):7505-7509. DOI:10.16965/ijar.2020.146

Experimental design

No comment

Validity of the findings

Use the ROC curve to find the specific cut-off points for at least the dimensions that show a statistically significant difference in males and females and/or the right and left MS.

Additional comments

In the legend for Figure 1 (b), the CT imaging plane is written wrongly as maxillary width measured in the axial image. Changes to be made also in the manuscript text line number 158 and 481

---

## Round 0.2 · Minor Revisions

Please address the remaining issues noted by reviewer 4.

Reviewer 3 ·

Basic reporting

no comment

Experimental design

no comment

Validity of the findings

no comment

Additional comments

no comment

Reviewer 4 ·

Basic reporting

No comment

Experimental design

No comment

Validity of the findings

1. Units are to be added for all the diameters you calculated and the volume measurement.
2. In the text file Figure 1 (ii) rightly mentions, the width measured in the Coronal view. But Legend for Figure 1 is still written as width measured in axial view.

---

## Round 0.3 · accepted · Accept

Thank you for the revision and congratulations! I have assessed the revision myself, and I confirm that the manuscript is ready for publication.